# Safety Assessment of Tunnel Portals for Site Selection Based on Spatial Information Geoprocessing

**Iau-Teh Wang** 

Department of Civil Engineering, R.O.C. Military Academy, Kaohsiung 83059, Taiwan; itwangroc@gmail.com

**Abstract:** The evaluation of portal locations for mountain tunnels is among the most crucial considerations during route selection and structural layout planning. The development of spatial information technology has provided a more objective approach for assessing the slope stability of potential portal sites. The simulations in such studies have been performed to evaluate potential hazards and slope stability. However, potential instabilities resulting from excavation are seldom considered in these studies. Therefore, a method based on spatial information technology was developed in this study for considering the potential impact of the direction and depth of excavations on portal stability. An analysis method for an infinite slope was integrated into the geographical information system for evaluating the stability of critical wedges. The proposed method provides a reasonable estimation comparable with that provided by the conventional slice method. The results of applying this method to six mountain tunnel portals where slope instability occurred during construction indicate that the actual outcomes agreed with the predicted outcomes. For potential portal site evaluation, the proposed method facilitates the rapid estimation of safety factors for various slope designations, which is useful for site selection.

**Keywords:** site selection; safety factors; slope stability; tunnel portal location; limit equilibrium method; potential hazards assessment

---

## 1. Introduction

Portal sections connect the inside and outside of tunnels. As shown in Figure 1, a portal section is typically considered a boundary location with 1–2 D (D is the tunnel diameter) of coverage from the entrance, although it varies depending on geological conditions [1–3]. A portal section is a sensitive environment typically located on a gentle slope with a thin-covering rock layer composed of weather-resistant materials. Portal stability is markedly influenced by geographical, geological, meteorological, and hydrological factors [4–8]. Wang and Huane [9] studied tunnel constructions in Taiwan and reported that landslides and slope collapses can be detrimental to the success of portal constructions. Therefore, selecting appropriate portal locations has become crucial in planning and designing mountain tunnels.

The slope at the entrance may be in a natural state of equilibrium as a result of long-term exposure to external forces such as weathering, erosion, and mass movement. However, excavation works following the commencement of a project can alter the original topography, thereby unbalancing the equilibrium and inducing landslides. Simultaneously, slope collapses typically occur as a result of insufficient coverage. Reasons for the aforementioned situations primarily depend on the relationship between the slip surface and the portal section. Even where tunnel portals are not proximal to the location of landslides, excavation work may lead to fractures in the stratum, thereby allowing rainwater or underground water to seep into the stratum, which can lead to project failure. Therefore, slope stability is a key factor for successful portal section construction.

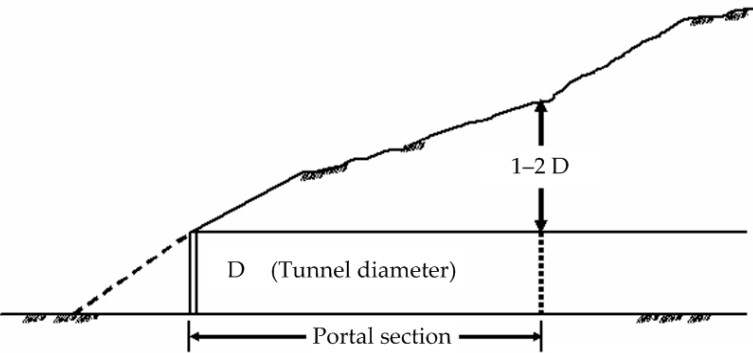

**Figure 1.** Schematic diagram for standard portal section ranges. The portal section varies depending on the geological conditions.

Entrance slope stability is a key consideration during portal location planning and related construction design. Approaches for evaluating slope stability can be classified as qualitative or quantitative. Based on site investigation and survey results, qualitative evaluations utilize expert and engineer experiences to assess the characteristics of a portal area. Quantitative evaluation involves calculating the slope stability as a function of force or moment equilibrium, and the limit equilibrium method (LEM) is most frequently used for such analyses. The LEM incorporates other methods in addition to the conventional slice method, which is well-established. The LEM method first involves determining the critical slope section for conducting stability analysis, and representative parameters are subsequently obtained through field or laboratory tests.

Previously, preliminary planning for site selection was undertaken with the assistance of experienced engineers. Initial site selection was established through field reconnaissance to obtain information for conducting subsequent analyses and design processes. This typically requires considerable manpower and is a highly time-consuming task. To conserve time and manpower, tools such as aerial photographs and geological and topographic maps are currently employed for preliminary site interpretation, to predict potential hazards and avoid unstable geological regions while selecting potential actionable engineering sites. Recently, the rapid development of geographical information system (GIS) technology has enabled the precise evaluation of geologic hazards [10–13]. GIS technology also provides an effective analytical tool for the large-scale assessment of geographical information to conduct rapid spatial data analysis [14,15]. Favorable results have been attained by studies in which GIS data were used to predict specific slope failure incidents [16]. The increased availability of GIS data in tunneling industries also provides strong functionality in data processing and analyses [17]. However, these studies have seldom considered potential instabilities resulting from excavation.

We considered incorporating the excavation depth and sliding range into mechanical analysis for calculating the entrance slope safety factor (*FS*) after commencing excavations, and subsequently applied this method for the preliminary assessment of location selection. By rapidly evaluating potential portal sites, this study provides a GIS-based method that integrates the slice method, which is potentially most widely used for estimating slope stability during the planning stage. Based on specific assumptions regarding the excavation depth and the potential slip surface, the safety factor of a specific location can be calculated by referring to various excavation orientations. The minimal safety factor can then be applied and considered a critical value for evaluating stability.

The following section introduces the proposed methodology to facilitate rapid selection of potential sites for mountain tunnel portals. Subsequently, six portals are examined to validate the proposed method. The influence of crucial assumptions is also addressed by comparing the proposed method with the conventional slice method.

## 2. Research Methods

A method for selecting a tunnel portal site was developed based on spatial information concepts. The terrain of target areas, excavation depth, direction, sliding range, and mechanical properties of geologic materials are considered.

### 2.1. Slope-Cutting Ambit and Critical Excavation Depth of Portals

For a specific area, the slope-cutting ambit for a portal is influenced by the relative entrance elevation and the slope (vertical vs. horizontal) used for cutting. The slope-cutting ambit is related to the critical excavation depth, the relative height from the cutting point height to the elevated portal entrance. To construct stable mountain portals and reduce the environmental impact, it is recommended to minimize landslides induced by excavation.

The cutting ambit of slopes is considered for evaluating potential portal sites, and the critical excavation depth is also considered in the proposed GIS-based method. For a typical double-lane highway tunnel with a width and height of 10–12 and 6–7 m, respectively, the critical excavation depth is approximately 10–15 m [18]. Thus, the initial value for calculating the slope stability was set at 15 m, and this value can be adjusted to fit dynamic field conditions if necessary. The excavation height of each stage was 5 m, as shown in Figure 2.

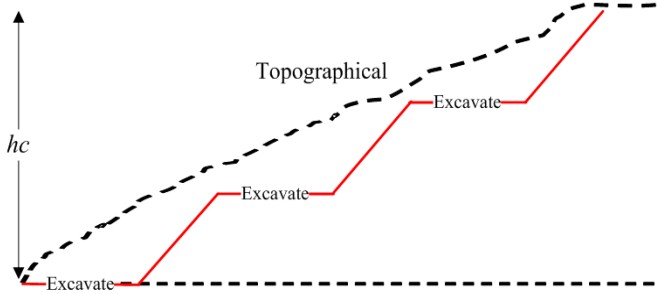

**Figure 2.** Schematic diagram for excavation slope of portal section.

If the critical excavation depth is known, the slice method is used to conduct a series of calculations to determine the potential sliding surface with a minimal safety factor for the cutting slope. The potential sliding mass size is typically proportional to the excavation height ($hc$). For simplicity, this study assumed that the sliding surface was planar and that it passed the slope toe. The distance between the highest cutting point and the crown of the sliding mass $d$ is proportional to the excavation depth. The slip area of the slope is $d = \tan\beta \times hc$. The $\beta$ can be arbitrarily specified to meet field conditions, and assumed $\tan\beta = \tan 45° = 1$ for the initial calculation performed in this study, as shown in Figure 3.

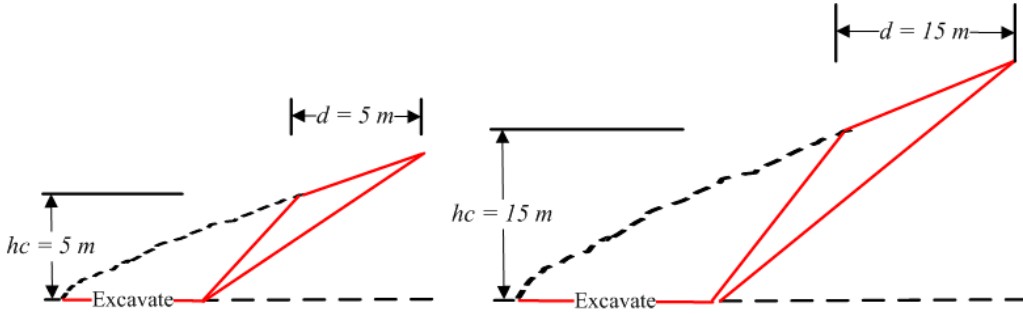

**Figure 3.** Schematic diagram for slip surface of slope excavation.

For a potential site, the portal direction can be planned to include sites with favorable slope stability. Every possible cutting direction should be assessed by considering topography. In this study, the safety factor of cut slopes was assessed for eight directions with a 45° interval (north, northeast, east, southeast, south, southwest, west, and northwest), as shown in Figure 4. The safety factor of the eight excavation directions was calculated, and the lowest safety factor was then applied as the representative safety factor.

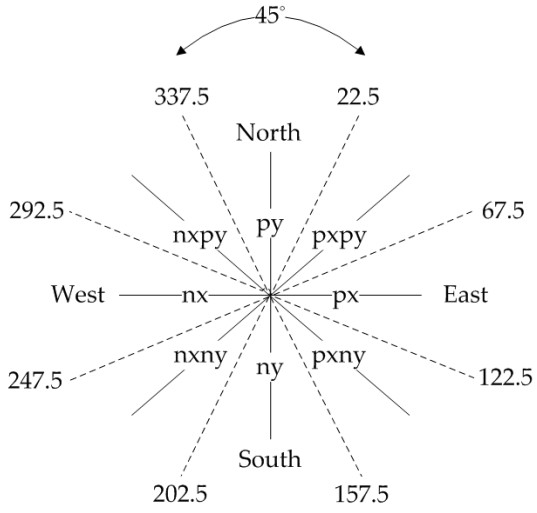

**Figure 4.** Schematic diagram of aspects of slope excavation at portal. The cutting slope safety factor (*FS*) in eight different directions. Each assessing area was set at every 45° interval and was divided as a slope aspect, clockwise. p symbolizes positive and n symbolizes negative. x is x-axis and y is y-axis.

## 2.2. Slope Stability Evaluation Using the Limit Equilibrium Method

For analyzing the slope stability of a potential site relative to the critical excavation depth and potential planar sliding surface, the LEM considers gravity and motion. For all shear-type failures, the geological material can be assumed to follow the Mohr–Coulomb failure criterion, in which the shear strength is a function of cohesion and the friction angle.

The *FS* for a potential sliding wedge (Figure 5) can be analyzed using the ratio between the driving force acting on the sliding surface $f_d$ and the resistant force of the plane $f_r$. The *FS* calculation involves resolving $f_d$ into action components parallel and perpendicular to the sliding surface. In other words, the angle between the failure plane and the horizontal plane $\theta$, the total length of the sliding surface *L*, and the weight of the wedge above the sliding surface *W* can be applied to obtain the resisting and driving forces on the sliding plane as follows:

$$f_r = W \times \cos\theta \times \tan\phi + cL, \tag{1}$$

$$f_d = W \times \sin\theta, \tag{2}$$

where c and $\phi$ are the apparent cohesion and friction angle of geomaterials, respectively.

The stability of the block shown in Figure 5 can be quantified based on the relationship between the resisting forces (*S*), normal forces (*N*) and weight of the sliding surface (*W*), which is known as the safety factor. Thus, the equation for the excavation can be expressed as:

$$S = N \times \tan\phi + cL, \tag{3}$$

$$\overline{AB} = \frac{hc}{\cos(90° - \beta)}, \tag{4}$$

$$\overline{BE} = \overline{AB}\sin(\beta - \theta), \tag{5}$$

$$L = \overline{AF} = \overline{AB}\cos(\beta - \theta) + \overline{BE}\cot(\delta), \tag{6}$$

$$W = \gamma \times \Delta ABF = \frac{1}{2} \times \gamma \times \overline{AF} \times \overline{BE}, \tag{7}$$

$$FS = \frac{f_r}{f_d} = \frac{W \times \cos\theta \times \tan\phi + cL}{W \times \sin\theta} \tag{8}$$

where $\gamma$ is dry unit weight of geomaterials.

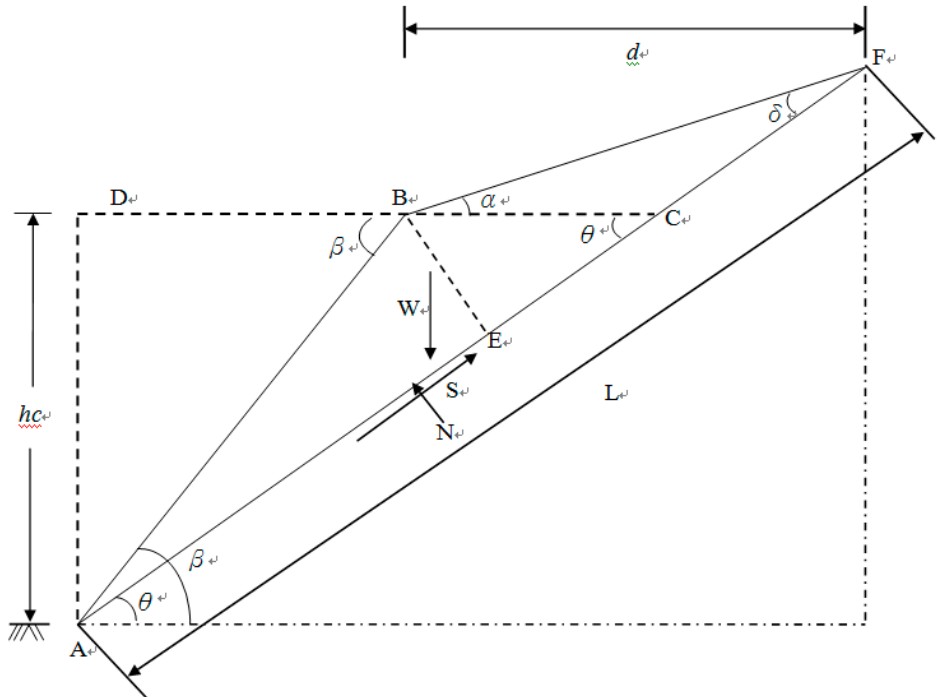

**Figure 5.** Schematic free body diagram for the assumed analyzed wedge based on the critical excavation depth and the potential planar sliding surface. The derived equation of the potential sliding wedge can be quantified by the relation between the resisting and driving forces, which is termed the *FS*.

Figure 6 shows a flowchart of the proposed methodology. The limit equilibrium analysis was coupled with a GIS. Based on the LEM, all factors employed in this study were transformed into grid spatial data using the GIS. A spatial model and a digital terrain model (DTM) in the GIS were employed to identify the excavation direction and dip inclination. The safety factor distribution map were converted into the vector format with the descriptions of types and units linked to the map using the built-in relational database management system in the GIS software. Geometric factors of slope, such as gradient and aspect, were calculated based on a DTM for Hualien, Taiwan, provided by the Institute of Aerial Survey for Agriculture and Forestry of Taiwan. The resolution of the DTM grid is $5 \times 5$ m.

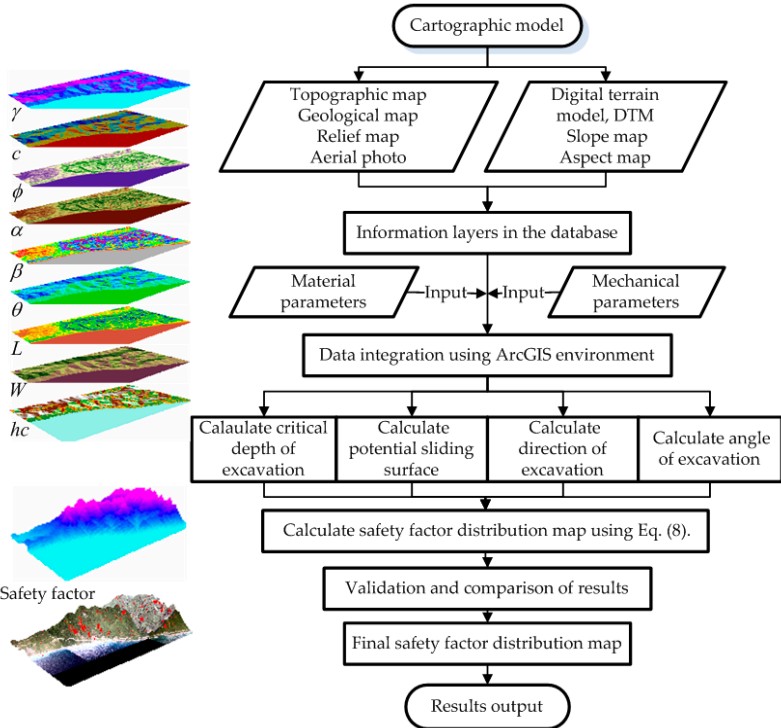

**Figure 6.** Flowchart showing the various phases of the performed methodology and all of the factors that were transformed into a grid-type spatial database. The safety factor distribution map and the description of types and units were stored in tables linked to the map using the built in relation data base management system of the GIS.

## 3. Application of the Proposed Method

To validate the proposed method, the slope stabilities of six portals constructed during a tunneling project were tested. The study area is located in Eastern Taiwan (Figure 7). The topography is highly variable, and south-facing slopes tend to be steeper than other slopes. Three tunnels (Tunnels 1, 2, and 3) were constructed for a highway renovation project. Among the six portals of the three tunnels, the south portal of Tunnel 2 and the south and north portals of Tunnel 3 suffered from slope instabilities during construction.

Figure 8 shows the geological map of the study area. The main stratum units include the Tuluanshan formation, Shuilian Conglomerate formation, Fanshuliao formation, Paliwan formation, and alluvial deposit. Table 1 lists the mechanical parameters and average unit weights of these formations. The average unit weight was back-calculated from actual landslide areas in the study area and from laboratory test results [15]. The DTM was used to calculate the distribution condition of the gradient and aspect of the slope. Subsequently, based on the grid resolution, the parameters of Equation (8) and safety factors were calculated using a database management system in the GIS software.

**Table 1.** Mechanical parameters for distinct formation in study area.

| Formation | Rock/Soil Type | $c$ (kPa) | $\phi$ (°) | $\gamma$ (kN/m$^3$) |
|---|---|---|---|---|
| Alluvial Deposit | gravel, sand, clay | 8.8 | 20 | 15 |
| Fanshuliao | shale | 11 | 25 | 21.62 |
| Paliwan | conglomerate | 15 | 30 | 23.35 |
| Tuluanshan | conglomerate | 10 | 35 | 22.13 |
| Shuilian Conglomerate | conglomerate | 24 | 29 | 22.78 |

The Tuluanshan formation is volcanic, formed from pyroclastic deposits. The lithology of this formation varies considerably, although it primarily comprises andesite gravel, tuff, and tuffaceous sediments. The Shuilian Conglomerate formation is mainly composed of marine conglomerate and sandstone. The Fanshuliao formation primarily consists of marine mudstone, sand, and interbedded shale, with a bottom layer of mudstone tens of meters thick, and an upper portion comprising sand and interbedded shale hundreds of meters thick. The Paliwan formation consists of conglomerate, mudstone, blue-gray sand, and interbedded shale. The alluvial deposits are Quaternary fluvial sediments.

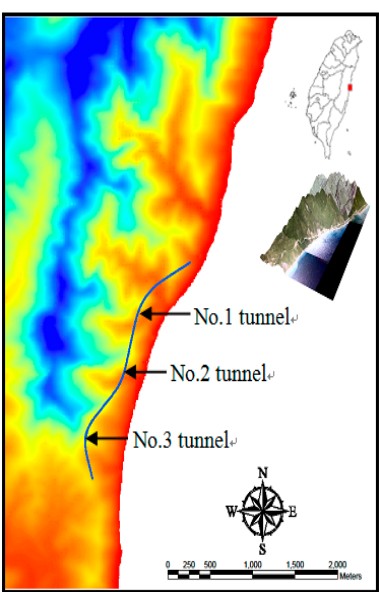

**Figure 7.** Study area located in the southeast area of Hualien, Taiwan.

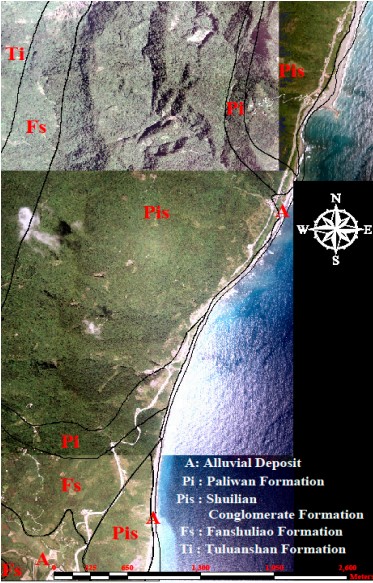

**Figure 8.** Geological distribution map of the study area.

Table 2 summarizes the results of the stability analysis of the excavation slope at the portal section. The vertical-to-horizontal ratio of the cutting slope is the most critical factor affecting the stability of the portal. If the ratio of the cutting slope is 1:1 (vertical:horizontal), the cutting slopes of all the six portals can be considered stable. However, if the cutting slope ratio for the south portal area of Tunnel 2 and both the south and north portals of Tunnel 3 is 1:0.5, their safety factors are less than 1.0. The north portal of Tunnel 2 becomes also unstable if the ratio is higher than 1:0.3. Therefore,

excavating these portal slopes may lead to instability. The cutting slope ratio is approximately 1:0.3 for both portals of Tunnel 2, and approximately 1:0.5 for both portals of Tunnel 3. The analytical results agree with the construction outcomes of the three portals. The difference in the north portal of Tunnel 2 may have resulted from underestimating the effect of the slope protection measures. For example, the terrain of target areas may have been covered with thin weather-resistant materials. Based on the results of this case study, the mechanical properties of geologic materials and the excavation gradient of slopes surrounding portals were considered to be key factors controlling the slope safety near the tunnel portal.

**Table 2.** Analysis results for stability of excavation slope at portal section.

| Tunnel | Portal Site | Vertical: Horizontal | | |
|---|---|---|---|---|
| | | **1.0: 1.0** | **1.0: 0.5** | **1.0: 0.3** |
| No. 1 | North | $FS \geq 1$ | $FS \geq 1$ | $FS \geq 1$ |
| | South | $FS \geq 1$ | $FS \geq 1$ | $FS \geq 1$ |
| No. 2 | North | $FS \geq 1$ | $FS \geq 1$ | $FS < 1$ |
| | South | $FS \geq 1$ | $FS < 1$ | $FS < 1$ |
| No. 3 | North | $FS \geq 1$ | $FS < 1$ | $FS < 1$ |
| | South | $FS \geq 1$ | $FS < 1$ | $FS < 1$ |

## 4. Discussion

Numerous factors affect slope stability, including terrain, underground water level, mechanical parameters, and the unit weight of geomaterials. For rapid evaluation of slope stability, performing geodetic or geological investigations is unrealistic. To evaluate the assumptions involved in the proposed fast method, the safety factors obtained through the proposed method and conventional slice method are discussed here.

### 4.1. Obtained Safety Factors

To elucidate the influence of the safety factor of a slope on the excavation depth and slip surface, we compared the results obtained from Equation (3) with the safety factor obtained using the STABL computer program to confirm the applicability of the proposed model. STABL is a computer program written for the general solution of slope-stability problems using a two-dimensional LEM. This program is used to perform slope stability analysis. The slip surface is considered as Bishop methods series of slope stability programs [19]. Three cutting slope ratios, 1.0:1.0, 1.0:0.5, and 1.0:0.3 (vertical: horizontal), were tested. The geomaterial parameters were $c = 25$ kPa, $\phi = 30°$, $\gamma = 13.55$ kN/m$^3$, $\gamma_{sub} = 20.50$ kN/m$^3$ (submerged unit weight), and $\gamma_{sat} = 25.36$ kN/m$^3$ (saturated unit weight). For comparison, the conventional slice method and the failure surface provided by the Bishop method of the STABL program were used to determine the minimal safety factor.

Figure 9 shows the safety factor obtained using the proposed method and conventional slice method at various excavation depths. The safety factors estimated using both methods decreased slightly as the critical excavation depth increased. The safety factors estimated using the proposed method were similar to those obtained using the conventional slice method when the cutting slope ratio was 1.0:0.5, but they were slightly higher when the cutting slope ratios were 1.0:1.0 and 1.0:0.3. Further research indicated that the variations were the result of various failure surfaces. When the cutting slope was gentle (i.e., the ratio was 1.0:1.0), the failure surface used in the slice method typically passed under the slope toe into a base failure type, thereby resulting in a lower safety factor. When the cutting slope was steep (i.e., the ratio was 1.0:0.3), the failure surface used in the slice method typically passed the cutting surface with a smaller failure circle, which also resulted in a lower safety factor. The analytical results are consistent with those reported by Wang and Huane [9], and they complied with those reported by the relevant literature [18], thereby validating the accuracy and applicability of the

proposed method. Therefore, the proposed method provides an efficient and rapid analytical tool for evaluating potential portal sites.

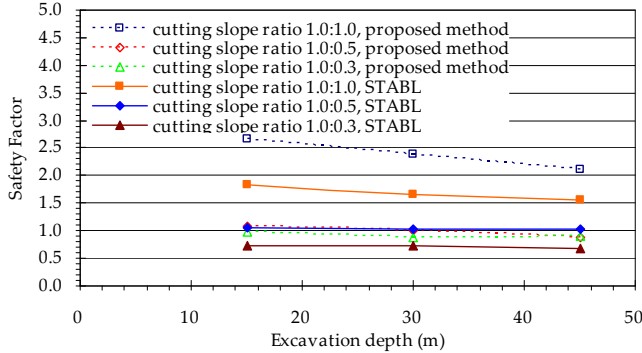

**Figure 9.** Safety factor obtained using the proposed method and the conventional slice method for various excavation depths.

### 4.2. Influence of Topography

Because of varying terrain features, the portal section of a tunnel is typically located on a slope. Any change in altitude during tunneling directly affects the slope gradient and height at the portal. Moreover, the selection of tunnel routes and construction planning are directly affected. At a specific location, the tunnel entrance topography influences both the portal slope-cutting ambit and slope safety factor. To select a potential portal site, existing terrain should be examined and tunnel entrance topography should be considered. The influence of the tunnel entrance topography is further discussed as follows.

Figure 10 shows the influence of the tunnel entrance elevation with a cutting slope ratio of 1.0:1.0 in the vicinity of the southern portal of Tunnel 3. The tunnel entrance elevation is 95 m, and the safety factors in most areas range from 2.0–3.0 (Figure 10b). When the tunnel entrance elevation was increased by 5 m (i.e., to 100 m), the safety factors decreased to 1.5–2.0 (Figure 10c). When the tunnel entrance elevation was lowered by 5 m (i.e., to 90 m), the safety factors increased to more than 3.0 (Figure 10a).

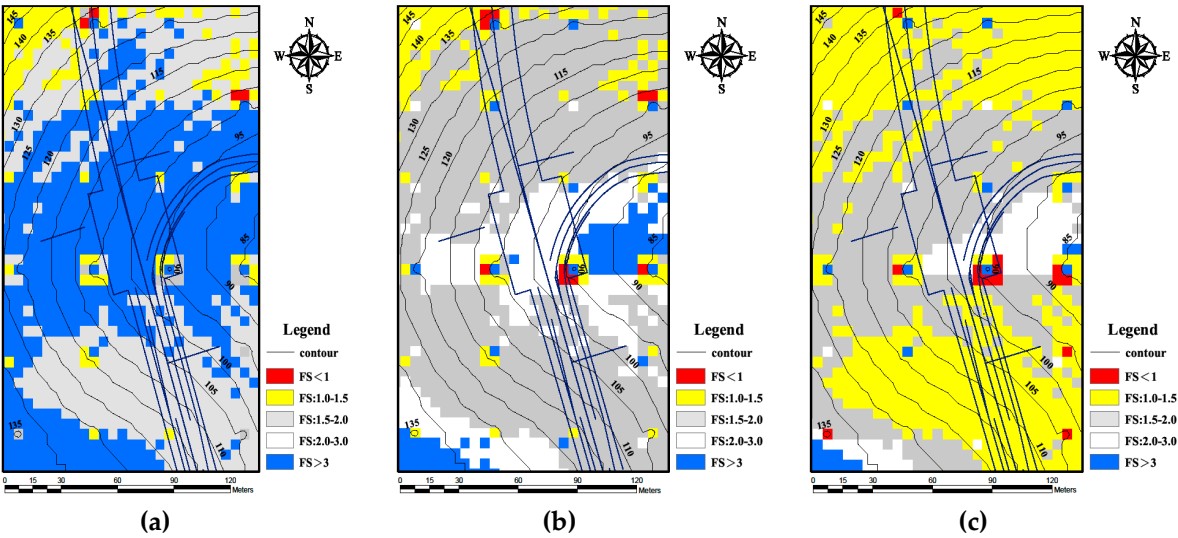

**(a)**　　　　　　　　　　**(b)**　　　　　　　　　　**(c)**

**Figure 10.** Influence of the tunnel entrance elevation on the safety factor of slopes with cutting slope ratio 1.0:1.0. (**a**) Portal locations elevation of 90 m, *FS* > 3. (**b**) Portal locations elevation of 95 m, *FS*: 2.0–3.0. (**c**) Portal locations elevation of 100 m, *FS*: 1.5–2.0.

Figure 11 shows the influence of the tunnel entrance elevation with a cutting slope ratio of 1.0:0.3. The tunnel entrance elevation is 95 m, and the safety factors in most areas range from 1.0 to 1.5 (Figure 11b). When the tunnel entrance elevation was increased or reduced by 5 m (i.e., to 100 or 90 m), the safety factors changed to 2.0–3.0 and less than 1.0, respectively (Figure 11a,c). The tunnel entrance elevation clearly exerted a substantial effect on the slope cutting safety factor. Determining an appropriate elevation is as critical as determining the tunnel route layout. This discussion verifies that the proposed method can be used to estimate the influence of the tunnel entrance elevation rapidly and effectively.

In this research, a method of tunnel portal site selection is developed based on the concepts of the GIS, and the suitable site of the tunnel portal could be evaluated with objective spatial information analyses techniques. The terrain of target areas, mechanical properties of geologic materials, and excavation gradient of slopes surrounding the portal, and the orientation of the portal of the tunnel are also taken into consideration by this method. According to the results of the case study and analyses of the safety factors, the geologic materials and the excavated gradient of slopes nearby the portals of tunnels are the key factors controlling the slope safety surrounding the portals of tunnels.

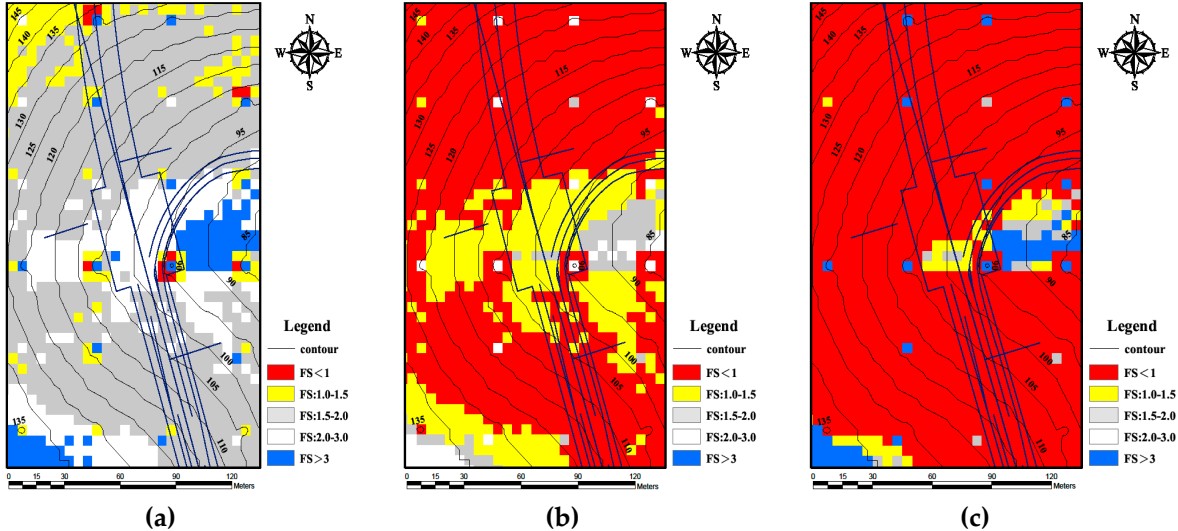

**Figure 11.** Influence of tunnel entrance elevation on the safety factor of slopes with cutting slope ratio 1.0:0.3. (**a**) Portal locations elevation of 90 m, *FS*: 2.0–3.0. (**b**) Portal locations elevation of 95 m, *FS*: 1.0–1.5. (**c**) Portal locations elevation of 100 m, *FS* < 1.

## 5. Conclusions

In this study, analysis technology was combined with the GIS software to improve portal section planning. Parameters such as unit weight, effective cohesion, and effective friction angle were selected as geological material parameters; gradient and aspect of slope were selected as slope geometric parameters, at the same time, a slip plane was assumed. Combining the parameters above, an assessment model involving limit equilibrium slope stability analysis linked with the GIS was developed for slope stability assessment at portal sections of a tunnel. After comparing landslide locations and scales between actual landslides and the results from the analysis model, which were derived in this study, the results for both were shown to coincide. As the results showed, the model of this study can act as the initial stage of analysis for the slope stability of a portal; the result provides good reference data for location selection of a portal and tunnel alignment.

**Funding:** This research received no external funding.

**Acknowledgments:** Author thanks the engineering advisory group of the Ministry of National Defense in Southern Taiwan, R.O.C., for assisting in this study.

**Conflicts of Interest:** The author declares no conflict of interest.

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
