# Peer review of "Safety Assessment of Tunnel Portals for Site Selection Based on Spatial Information Geoprocessing"

_infrastructures, doi:10.3390/infrastructures4040070_

Round 1

Reviewer 1 Report

The paper describes a methodology for preliminary assessment of potential landslides induced by excavations near tunnel portals. The methodology uses geographical information system (GIS) to provide geometrical data of terrain which combined with material properties of geomaterials give the base for assessing the factor of safety (FS) at each point of the GIS grid. In principle, such methodology might be a valuable tool for infrastructure engineers during the stage of route planning.

The limit equilibrium method (LEM)  with linear sliding surface was chosen by the author as a method for computing the factor of safety. Although the method oversimplifies the problem of slope stability by assuming the sliding wedge to be a rigid body and the distribution of the contact stress along the sliding surface to be uniform, it is still appropriate for preliminary check of slope sliding hazard.

Nevertheless, the application of LEM as described in the paper is confusing and introduces further simplifications that are not justified. The parameter “beta” is defined to be a ratio of “d” over “hc” and assumed to be equal to 1. It is not explained why this assumption was made. Moreover, the distance “d” is denoted in the figure, but the distance “hc” is not. In the same figure and in later equations there is an angle also denoted by “beta” which is very confusing. Another confusing information is in equation (6) where the distance between points B and C is assumed to be 0.5*d which does not hold in general. The fact that this does not hold is clearly visible directly on the figure 3 where the dimension denoted as 0.5*d is longer than half of dimension d.

To make the paper useful, the method for computing FS has to be described precisely so that the interested reader is able to implement the computations on his/her own. The given parameters such as the inclination of the terrain at certain point and parameters of the Mohr-Coulomb should be stated first. Then the parameters of the excavation (depth, inclination) should be specified. Finally, the other input parameters, such as the inclination of the potential slide surface, should be either assumed or optimized. All the input parameters that has a geometrical meaning has to be present in figure similar to figure 3. Only after all the input parameters were specified, the equations for computing FS should be presented.

The remaining parts of the paper describe the model validation and compares the prediction of slope sliding hazard with real cases. The factor of safety computed by the presented variant of LEM method is also compared to values obtained by STABL software. Nevertheless, the method on which the software is based is not mentioned and the software is not even cited in the references.

To summarize my opinion, the idea of the paper - to assess slope sliding hazard induced by excavation near tunnel portal directly from GIS - is interesting and potentially useful but due to the significant flaws described above the paper can be published only after major revision and another round of review.

Author Response

The author is grateful for reviewer kind suggestion.

Reviewer 2 Report

The article presents a method that automates in a GIS environment the calculations of several parameters that evaluate safety of tunnel portals.

The article presents multiple weaknesses from a scientific point of view.

The qualitative methodology usually applied in this type of work is indicated and a quantitative methodology is proposed based on the results of a set of calculations that are systematically implemented and applied to different tunnel portals orientations.

The holistic methodology proposed is based on the Limit Equilibrium Method (LEM) considered and described, for which it is necessary to provide different data sources that characterize the terrain: geometry, topography and geology.

The main weaknesses are:
The problem or the initial hypothesis that motivates the development of a new methodology is not formally specified.
A sufficient state of the art is not provided. References 4-8 and 9 that appear on lines 31 and 32 are provided by the first of the references used in the article (Chang & Liu, 1999). The second paragraph of the introduction is not justified (there are no references that cement the arguments).

The method is not adequately described. The formulation is presented and some description is duplicated between the text and the captions, but some additional details or descriptions of the motivations / reasoning to use these calculations are missing.

The description of the implementation in the form of a flowchart is sufficient for a general description, but no implementation details are provided. It follows that it is in ArcGIS but not how it has been implemented.

It is not indicated how the parameters of the geological compounds that appear in each case study are incorporated into the calculations.

Figures 5 and 6 should be more readable.

Results are compared in the discussion section with those obtained with the STABL program, which should have been cited or referenced from the beginning as an available tool to make some kind of calculation related to the object of the methodology presented. This program is also not described. It should be done in the introduction or in the state of the art.

The discussion sections and conclusions are unclear, cutting edge comparisons are made continuously at intervals without qualitatively valuing them, or it is about displaying results graphically like figures 8 and 9 that do not properly communicate what they are representing, it is not interpreted properly what the tunnel portal.
The conclusions should be related to the initial problem, how the proposed quantitative method improves the results versus the qualitative ones based on experience or explorations.

Author Response

(The authors gave the same response as above.)

Reviewer 3 Report

This submission is a very interesting paper with a very applied topic and falls in the scope of the journal.Although, it has some potential for improvement.

Introduction

The general description of the problem (Introduction) and the description of its importance for the science and the society could be further improved. The degree of innovativeness of the methodological approach is not convincingly demonstrated. Some more details about its innovative features could further improve the quality of this paper. Why is this paper likely to be cited in the future?

Method

A bit more text regarding the originality of this work and why it contains new results that significantly advance the research field.

Results

I believe that adding a bit more text on why the results of the method are satisfactory (evaluation approach) will increase the quality of this work Could the results be more satisfactory if you have changed something in the methodology? Are the results (or the method) sensitive to this specific study area?

Discussion

In the Discussion section I would have wished to see more information on the actual meaning of the findings and how the results add to the broader topic as well as the specific scientific field

Conclusion

The "Conclusions" section, could be further improved by describing the importance of this work, the highlight of potential further development of this methodology.

Author Response

(The authors gave the same response as above.)

Round 2

Reviewer 1 Report

I went through the modified parts of the article and the author’s comments but the response 3 unfortunately does not address my previous suggestions. In particular, my comments in the first round of review were:

Point 3: Nevertheless, the application of LEM as described in the paper is confusing and introduces further simplifications that are not justified. The parameter “beta” is defined to be a ratio of “d” over “hc” and assumed to be equal to 1. It is not explained why this assumption was made. Moreover, the distance “d” is denoted in the figure, but the distance “hc” is not. In the same figure and in later equations there is an angle also denoted by “beta” which is very confusing. Another confusing information is in equation (6) where the distance between points B and C is assumed to be 0.5*d which does not hold in general. The fact that this does not hold is clearly visible directly on the figure 3 where the dimension denoted as 0.5*d is longer than half of dimension d.

The highlighted points are not fixed or explained in the revised manuscript. There are still two different quantities denoted with beta. The paper still wrongly assumes that the distance BC is 0.5*d which is not generally correct. 

Also, as I previously stated

..the method for computing FS has to be described precisely so that the interested reader is able to implement the computations on his/her own. The given parameters such as the inclination of the terrain at certain point and parameters of the Mohr-Coulomb should be stated first. Then the parameters of the excavation (depth, inclination) should be specified. Finally, the other input parameters, such as the inclination of the potential slide surface, should be either assumed or optimized. All the input parameters that has a geometrical meaning has to be present in figure similar to figure 3. Only after all the input parameters were specified, the equations for computing FS should be presented.

This has to be clearly explained in the final manuscript. First, specify all parameters controlling the geometry of the sliding mass, for example: “hc - excavation depth, beta - angle of excavation face, d - horizontal distance of top point of excavation face B and upper point of sliding surface D, we chose d=hc”, etc. Then list the material parameters entering the calculation. For example: “only angle of internal friction phi, cohesion c and specific weight gamma enter the calculation FS”. Then list all the equation that you are using to compute FS.

This part might be in paper body or in appendix.

Author Response

(The authors gave the same response as above.)

Reviewer 2 Report

The authors have improved the article thanks to the comments of the reviewers.
In the case that corresponds to me, some explanations, figures have been improved and the conclusions have been modified.

Author Response

(The authors gave the same response as above.)
